# ORTHCAPS: AN ORTHOGONAL CAPSNET WITH SPARSE ATTENTION ROUTING AND PRUNING

## ABSTRACT

Redundancy is a persistent challenge in Capsule Networks (CapsNet), leading to high computational costs and parameter counts (Jeong et al., 2019; Sharifi et al., 2021; Renzulli & Grangetto, 2022). Although previous works have introduced pruning after the initial capsule layer, dynamic routing's iterative and fully connected nature reintroduces inefficiencies and redundancy in deeper layers. In this paper, we propose the Orthogonal Capsule Network (OrthCaps) to reduce redundancy, improve routing performance and decrease parameter count. Specifically, an efficient pruned capsule layer is placed to discard redundant capsules and dynamic routing is replaced with orthogonal sparse attention routing. Besides, we orthogonalize weight matrices during routing to ensure feature diversity and sustain low capsule similarity, the idea of which is inspired by the application of orthogonality in Convolutional Neural Networks (CNNs). Moreover, a novel activation function named Capsule ReLU is proposed to address vanishing gradient. Our experiments on baseline datasets affirm the efficiency and robustness of OrthCaps in classification tasks, in which ablation studies validate the criticality of each component. Remarkably, with only 110k parameters, merely 1.25% of a standard Capsule Network's total, OrthCaps-Shallow outperforms state-of-the-art (SOTA) benchmarks on four datasets, while OrthCaps-Deep attains nearly SOTA accuracy with 1.2% of its parameters on four datasets. The code is available at ornamentt/Orthogonal-Capsnet (github.com).

## 1 INTRODUCTION

CapsNet replaces neurons with capsule vectors, where the capsule length denotes the existence probability of entities in the image, and its direction indicates captured features(Sabour et al., 2017). Thus, a high similarity of the two capsules' directions implies they extract analogous features. Recent studies have mentioned that Capsnet contains redundant capsules (Chen et al., 2022; Sharifi et al., 2021; Renzulli & Grangetto, 2022). As evidence, Figure (1) shows 48.2% of primary capsule pairs exhibit cosine similarities above 0.65, indicating significant redundancy.

While some works have employed pruning techniques at the primary capsule layer(Renzulli et al., 2022), deeper layers continue to exhibit high similarity issues, as illustrated in Figure (6). This deep redundancy is primarily a result of dynamic routing. In this mechanism, every lower-level capsule connects to all higher-level ones. This full connection structure leads to potential transmission of redundant information. Furthermore, weight matrices in routing can shift capsule directions, increasing the reduced capsule similarity after pruning and causing feature overlap. Such overlap not only impairs routing performance but also reintroduces redundancies in subsequent layers. This means that despite the initial pruning, new redundancies emerge in subsequent layers due to dynamic routing. Additionally, dynamic routing requires multiple iterations and repeatedly updating coupling coefficients until convergence, further straining computational resources.

Considering these challenges, and inspired by the introduction of orthogonality in CNNs to reduce filter overlaps, we integrate orthogonality into capsule networks. Our proposed solution, the Orthogonal Capsule Network (OrthCaps), addresses the iterative convergence, directional shifts in capsule vectors, and the fully connected structure of dynamic routing. We present two versions of OrthCaps: a lightweight model, OrthCaps-Shallow (**OrthCaps-S**), and an efficient deep model, OrthCaps-Deep (**OrthCaps-D**).

Firstly, we introduce a pruned capsule layer following the primary capsule layer, This layer eliminates redundant capsules, retaining only the essential and representative ones. The importance of capsules is gauged by $L_2$-norm, as it reflects the existence probability of the entities they represent. Given that a capsule's direction signifies the features it extracts, we measure their correlation using cosine similarity and employ broadcasting and matrix multiplication for algorithmic efficiency.

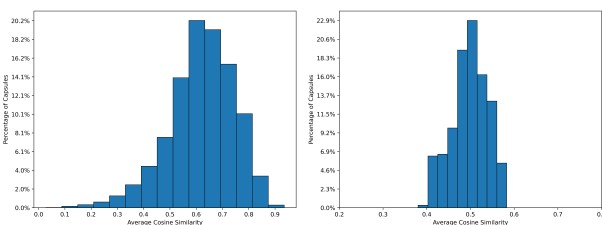

Figure 1: **Left:** In CapsNets primary capsule layer, 48.2% of capsule pairs have cosine similarities greater than 0.65, indicating significant redundancy among capsules. **Right:** After introducing the Pruned Layer, capsule similarities effectively decrease. (Detailed in Section 3.2)

Secondly, dynamic routing is replaced with attention routing, eliminating the need for iteration. For solving the fully-connected problem, We leverage sparsemax-based self-attention to produce an attention map, which selectively amplifies relevant feature groups corresponding to specific entities while downplaying irrelevant ones. For OrthCaps-S, a simplified attention-routing mechanism is adopted, optimizing parameter count and computational demands.

Thirdly, to address the issue of capsule vector direction shifts, we introduce orthogonality into capsule networks. An orthogonal weight matrix preserves the direction of capsule vectors, thus mitigating feature interference. Utilizing Householder orthogonal decomposition, we enforce orthogonality in the weight matrices during attention routing, which sustains low inter-capsule correlation and enriches feature diversity.

Lastly, we propose an activation function called Capsule ReLU, tailored for deep capsule networks. Although squash prevails in capsule networks, its saturation regions, akin to the sigmoid function, lead to vanishing gradient problems in deeper architectures. Thus, Capsule ReLU is designed to better suit OrthCaps-D.

**Contributions.** To summarize our work, we make the following contributions:

1) A novel orthogonal sparse attention routing mechanism is proposed to replace dynamic routing. Notably, it is the first time orthogonality has been introduced into capsule networks. This simple, penalty-free orthogonalization method is also adaptable to other neural network architectures.

2) A pruned capsule layer is placed to alleviate capsule redundancy and a new activation function named Capsule ReLU is proposed for deep capsule networks.

3) Two OrthCaps versions are created: OrthCaps-S and OrthCaps-D. OrthCaps-S sets a new benchmark in accuracy with just 1.25% of CapsNet's parameters on datasets of MNIST, SVHN, smallNORB, and CIFAR10. OrthCaps-D excels on CIFAR10, CIFAR100 and FashionMNIST while keeping parameters minimal.

## 2 RELATED WORK

**Capsule Neural Networks.** Dynamic routing was first introduced by (Sabour et al., 2017). Although numerous studies have leveraged attention strategies (Hoogi et al., 2019; Peng et al., 2020; Mazzia et al., 2021) to refine dynamic routing, the full connection structure and redundancy introduction seldom changes(Sabour et al., 2017). (Choi et al., 2019) incorporated attention into capsule routing via a non-iterative feed-forward operation. (Tsai et al., 2020) introduced parallel iterative routing, which did not address the complexity of iterative requirements. Furthermore, (Jeong et al., 2019; Sharifi et al., 2021; Renzulli et al., 2022) incorporated pruning, but did not account for new redundancies introduced by dynamic routing. (Jeong et al., 2019) introduced a ladder structure to CapsNet, using a pruning algorithm based on code vectors. (Sharifi et al., 2021) created a pruning layer based on Taylor Decomposition. (Renzulli et al., 2022) utilized LOBSTER to create a sparse parse tree. Different from existing research, this paper incorporates pruning, orthogonality and sparsity to effectively eliminate redundancy.

**Orthogonality.** Various methods were proposed to introduce orthogonality into neural networks, which can be categorized into hard and soft orthogonality. Hard orthogonality maintains matrix orthogonality throughout training by either optimizing over the Stiefel manifold (Li et al., 2020;

Huang et al., 2018), or parameterizing a subset of orthogonal matrices (Trockman & Kolter, 2021; Singla & Feizi, 2021; Virmaux & Scaman, 2018). These methods incur computational overhead and result in vanishing or exploding gradients. Soft orthogonality, on the other hand, employs a regularization term in the loss function to encourage orthogonality among column vectors of weight matrix without strict enforcement (Wang et al., 2020; Qi et al., 2020; Huang et al., 2020). Yet, strong regularization overshadows the primary task loss, while weak regularization fails to effectively encourage orthogonality. We leverage Householder orthogonal decomposition (Uhlig, 2001; Mathiasen et al., 2020) to achieve strict matrix orthogonality, minimizing computational complexity and obviating the need for additional regularization terms.

## 3 METHODOLOGY

### 3.1 OVERALL ARCHITECTURE

We introduce OrthCaps, offering both deep (OrthCaps-D) and shallow (OrthCaps-S) architectures to minimize parameter count while exploring the potential for deep multi-layer capsule networks. As illustrated in Figure (2)(a), OrthCaps-D comprises five key components: a convolutional layer, a primary capsule layer, a pruned capsule layer, capsule blocks and a flat capsule layer.

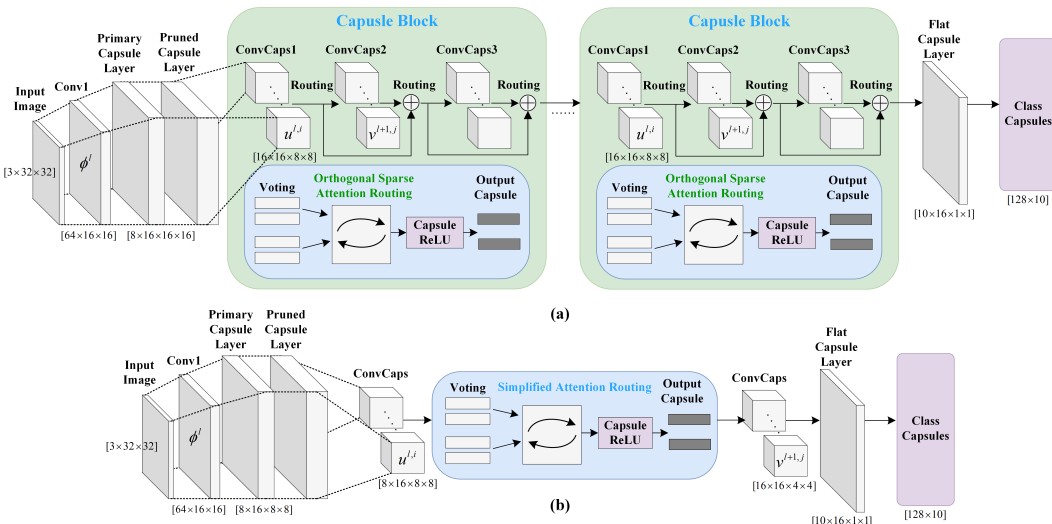

Figure 2: **(a):** In CIFAR10 classification task, the OrthCaps-D model comprises 7 capsule blocks, each with 3 capsule layers, interconnected via shortcut connections and orthogonal sparse attention routing. **(b):** The OrthCaps-S model contains two capsule layers coping with CIFAR10 and does not use any capsule layer with MNIST. These layers are linked through simplified attention routing.

Given an input image $x \in \mathbb{R}^{H \times W \times 3}$, low-level features $\Phi^l \in \mathbb{R}^{(B,C,W^l,H^l)}$ are extracted through four convolutional layers. The primary capsule layer generates initial capsules $u^l \in \mathbb{R}^{(B,n,d,W^l,H^l)}$ with a kernel size of 3 and stride of 2. A pruned capsule layer is then obtained to remove redundant capsules. Each capsule block contains three convolutional capsule layers with depthwise convolutions and shortcut connections preventing gradient vanishing. In capsule blocks, lower-level capsules $u^l$ are routed to the next layer $v^{l+1}$ via orthogonal sparse attention routing and Capsule ReLU. The block structure permits stacking to construct deeper capsule networks. The flatcaps layer comprises depthwise convolutional layers with a 3x3 kernel and a stride of 2 for capsule map reduction, and 1x1 pointwise convolutions with a stride of 1 for dimensionality mapping.

OrthCaps-S, as illustrated in Figure (2)(b), replaces the complete attention routing with a simplified version, retaining a single cell with adjustable convolutional capsule layers. Convolutional capsules in the primary layer utilize a 9x9 kernel with a stride of 1. Other layers and the activation function are consistent with OrthCaps-D.

## 3.2 Pruned Capsule layer

The generation of capsules starts with the primary capsule layer, which derives its input from feature maps of preceding convolutional layers. Reducing redundancy at this stage is crucial to ensure low-correlated capsules, allowing for efficient feature representation. To achieve this, we introduce an efficient capsule pruning algorithm after the primary capsule layer. The Algorithm 1 comprises the following steps:

**Capsule Sorting:** To ensure that the less important capsule is discarded when the similarity between a pair of capsules is high, each capsule $u_{l,i}$ is ranked based on its $L_2$-norm $\|u_{l,i}\|_2$. This norm indicates the existence probability of entities extracted by $u_{l,i}$, indicating the importance of capsule $u_{l,i}$. The sorted capsules are stored in tensor $u_{\text{sorted}}$. For the 5D tensor $[B, \text{num\_capsules}, \text{dim\_capsules}, W, H]$ of $u_l$, we reshape it to $[B, \text{num\_capsules}, \text{dim\_capsules} \times W \times H]$ to simplify computation.

**Redundancy Definition:** The direction of each capsule vector represents specific features. Capsules with closer directions indicate similar features and entities. Therefore, we utilize the cosine similarity of capsule directions to measure their correlation and redundancy. The similarity matrix $S$ for $u_{\text{sorted}}$ is computed using broadcasting.

**Pruning:** The capsule pair with similarity exceeding the threshold $\theta = 0.7$ is considered oversimilar. Then, the capsule with a lower rank in list $u_{\text{sorted}}$ of the pair is deemed redundant and deactivated by multiplying with a mask matrix $M$.

---

**Algorithm 1: Efficient Capsule Pruning**

**Require:** $u \in \mathbb{R}^{B \times n \times d \times W \times H}$, $\theta$
**Ensure:** $u_{\text{pruned}} \in \mathbb{R}^{B \times n \times d \times W \times H}$
1: Reshape $u \to u_{\text{flat}} \in \mathbb{R}^{B \times n \times (d \times W \times H)}$
2: Compute $L_2$-norm: $\|u_{\text{flat}}\|_2$
3: Sort capsules by $L_2$-norm: $u_{\text{sorted}}$
4: Flatten $u_{\text{sorted}}$ to $u_{\text{flat}} \in \mathbb{R}^{B \times n \times (d \times W \times H)}$
5: $S = \text{cosine\_similarity}(u_{\text{flat},i}, u_{\text{flat},j})$
6: Create mask $M$ where $S > \theta$
7: Prune using $M$: $u_{\text{pruned}} = u_{\text{sorted}} \odot M$
8: **return** $u_{\text{pruned}}$

---

## 3.3 Orthogonal Sparse Attention Routing

We introduce the orthogonal sparse attention routing to replace dynamic routing, which enable non-iterative and less redundant feature transmission from lower-level to higher-level capsules.

### 3.3.1 Routing Algorithm

Let $u_{l,i}$ and $v_{l+1,j}$ represent capsules at layer $l$ and $l + 1$ respectively, each with dimension $d$. We employ three weight matrices $W_Q, W_K, W_V \in \mathbb{R}^{d \times d}$ to derive keys, queries, and values from $u_{l,i}$. $Q = W_Q \times u_{l,i}, K = W_K \times u_{l,i}, V = W_V \times u_{l,i}$. Specifically, $W_Q, W_K$, and $W_V$ are designed as orthogonal matrices, enabling them to project capsule $u_{l,i}$ into a $d$-dimensional orthogonal subspace.

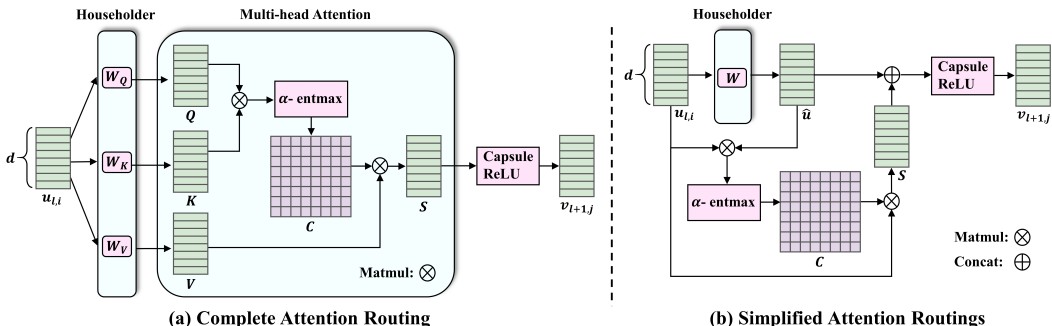

(a) Complete Attention Routing   (b) Simplified Attention Routings

Figure 3: Orthogonal self-attention routing.

As shown in Figure (3), attention routing aims to produce coupling coefficients $c_{ij}$, which quantifies the information transmitted from a lower-level to a higher-level capsule. The coupling coefficient matrix $C_{ij}$ is derived from the attention map $C$, generated through the dot product of queries and keys, $C = \alpha\text{-Entmax}(QK^T/\sqrt{d})$. Here, we replace the softmax function in the original attention mechanism with the $\alpha$-Entmax function (Peters et al., 2019) to enhance the sparsity of the attention

map, thereby encouraging routing to prioritize more important capsules while minimizing irrelevant information transfer. The vote $s_{i,j}$ is computed as the product of $V$ and $C$. Higher-level capsules $v_{l+1,j}$ are generated by $s_{i,j}$ from a multi-head self-attention mechanism with 16 heads, using the nonlinear activation function $g$.

$$v_{l+1,j} = g(s_{i,j}) = g(\text{Entmax}_{(\alpha)}(QK^T/\sqrt{d}) \times V) \tag{1}$$

For simplified attention-routing in Figure (3), we condense prediction matrices $W$ from three to one and replace $K, Q, V$ with $u_{l,i}$. $\hat{u}_{l,i}$ is the prediction for $v_{l+1,j}$. The attention map $C$ is obtained using $\alpha$-entmax with the dot product to produce the vote $s_{i,j} = u_{l,i} \times C = u_{l,i} \times (\text{Entmax}_{(\alpha)}(\hat{u}_{l,i}u_{l,i}^T/\sqrt{d}))$. $s_{i,j}$ is concatenated with $\hat{u}_{l,i}$ and then processed through $g$ to produce $v_{l+1,j}$. Notably, standard convolutions are supplanted by depthwise convolutions to minimize parameter count. Without any iteration, attention routing reduces computational complexity.

### 3.3.2 ORTHOGONALIZATION OF WEIGHT MATRIX

The pruned capsule layer diminishes capsule similarity to reduce redundancy. As we analyzed in Section 1, the weight matrix modifies the capsule vector's direction, potentially affecting the low correlation among internal capsules of $K, Q, V$, thus introducing new redundancy in subsequent layers. Orthogonal projection maintains the direction of vectors, thus preserving the low correlation and preventing feature overlap, which augments the performance of attention routing and pruning.

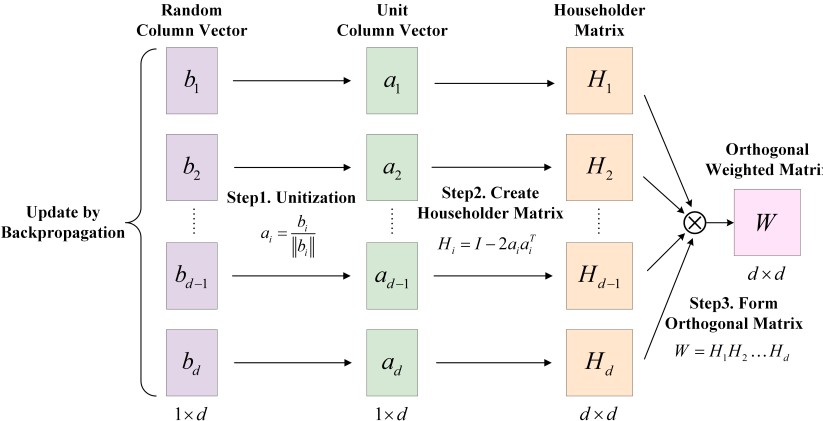

Figure 4: The computing process of HouseHolder orthogonalization method.

Let $W$ be the weight matrix requiring orthogonalization. As shown in Figure (4), Householder orthogonal decomposition theorem is employed to formulate an endogenously optimizable orthogonal matrix. The essence of this approach is in the following algebraic lemma (Uhlig, 2001):

**Lemma 1**: Any orthogonal $n \times n$ matrix is the product of at most $n$ orthogonal Householder transformations.

Let $d$ represent the dimension of the capsule. Based on Lemma 1, an orthogonal matrix $W_v \in \mathbb{R}^{d \times d}$ can be formulated in Equation (2):

$$W = H_0 H_1 \ldots H_{d-1} \tag{2}$$

Each $H_i$ represents a Householder transformation, defined as $H_i = I - 2a_i a_i^T$, where $a_i$ is a unit column vector. We utilize a set of randomly generated column vectors $\{b_i | i = 0, \ldots, d-1\}$ instead of $a_i$ to construct $H_i$ as detailed in Equation (3). During training, $b_i$ is optimized through gradient backpropagation. $W$ inherently preserves its orthogonality during training.

$$W = \prod_{i=0}^{d-1} \left( I - \frac{2b_i b_i^T}{\|b_i\|^2} \right) \tag{3}$$

**Lemma 2**: $W_Q, W_K$, and $W_V$ constructed using Equation (3) are Orthogonal.

Following Equation (3), $W_Q$, $W_K$, and $W_V$ could easily be orthogonalized, where the proof is provided in Appendix A.3.1. The proposed orthogonalization method for weight matrices is generalizable to any neural network, not limited to capsule networks. Householder orthogonalization enables computationally efficient transformation of arbitrary coefficient matrices into orthogonal matrices without any additional penalty terms in the loss function.

### 3.4 CAPSULE RELU

The activation function is an indispensable part of routing. However, as shown in Figure (8) in Appendix A.4.2, squash has saturation regions similar to the sigmoid function, which may result in vanishing gradients during backpropagation. Therefore, we incorporate a capsule structure with ReLU, which excludes a saturation region and avoids the vanishing gradients.

The squash function, $v_j = \frac{||s_j||^2}{1+||s_j||^2} \frac{s_j}{||s_j||}$, serves two primary functions: constraining the capsule length to interval $[0, 1]$ and preserving the capsule's direction. Replacing squash with ReLU directly would compromise these essential properties, leading to a large decline in network performance. To resolve this, we integrate ReLU with BatchNorm to compress the capsule length while maintaining its direction, as outlined in Equation (4):

$$v_j = ReLU\left(BatchNorm\left(||s_j||^2\right)\right)\frac{s_j}{||s_j||^2} \qquad (4)$$

In contrast to the neuron-level ReLU, Capsule ReLU performs group-level activation on capsules. If the $L_2$-norm of a capsule falls below zero after batch normalization, all elements within that capsule are zeroed out, thereby introducing sparsity.

## 4 EXPERIMENTS

### 4.1 EXPERIMENTAL SETUP

**Implementation details and datasets**

We implemented OrthCaps using PyTorch 1.12.1 on Python 3.9. For training, we adopted the margin loss as defined in (Sabour et al., 2017). We opted to exclude the reconstruction loss, observing minimal performance benefits in our experiments. Our model utilized the AdamW optimizer, combined with a cosine annealing learning rate scheduler and a 5-cycle linear warm-up. The initial configurations are learning rate at 5e-3, weight decay at 5e-4, and a batch size of 512. The training was facilitated by four GTX-3090 GPUs. We conducted experiments on SVHN(Netzer et al., 2011), smallNORB(LeCun et al., 2004), affNIST(Sabour et al., 2017), CIFAR10, and MNIST(LeCun, 1998) for OrthCaps-S. OrthCaps-D was trained and tested on CIFAR10, CIFAR100(Krizhevsky, 2009), Fashion-MNIST(Xiao et al., 2017), and MNIST. We resized SmallNORB from $96 \times 96$ to $64 \times 64$ and subsequently cropped it to $48 \times 48$, in line with (Sabour et al., 2017). All other datasets retained their original sizes, and data augmentation followed by (Hinton et al., 2018). To facilitate reproducibility, we have detailed the hyperparameters in Appendix A.2.

**Comparison baselines**

We benchmarked OrthCaps against various baseline models. For OrthCaps-S, we compared it with Efficient-Caps(Mazzia et al., 2021), CapsNet(Sabour et al., 2017), Matrix-CapsNet with EM routing(Hinton et al., 2018), AR CapsNet(Choi et al., 2019), AA-Capsnet(Pucci et al., 2021), DA-CapsNet(Huang & Zhou, 2020) and a standard 7-layer CNN. For OrthCaps-D, we used baselines such as CapsNet (7 ensembles), AR CapsNet (7 ensembles), RS-Capsnet(Yang et al., 2020), Inverted Dot-Product(Tsai et al., 2020), DeepCaps(Rajasegaran et al., 2019), ResNet-18(He et al., 2016), and VGG-16(Simonyan & Zisserman, 2014). All comparative results were derived from running official codes with the same hyperparameters in Appendix A.2.

### 4.2 CLASSIFICATION PERFORMANCE COMPARISON

Table (1) illustrates the classification performance of OrthCaps-S and OrthCaps-D, with model sizes denoted by Param and computational demands represented as FLOPS[M]. All models utilize a

backbone of 4 convolutional layers and undergo training for 300 epochs. The Param and FLOPS[M] of each table are tested on MNIST and CIFAR10, respectively. An asterisk (*) signifies that no official code is available, so we refer to the model performance stated in the original papers.

Table 1: **Top:** OrthCaps-S ranks as the top or second best across five datasets, standing out as being resource-efficient with only 105.5K parameters and 673.1M FLOPS. **Bottom:** OrthCaps-D shows competitive performance with fewer parameters and less computational cost.

| Shallow Networks | Param↓ | FLOPS[M]↓ | MNIST | SVHN | smallNORB | CIFAR10 |
|---|---|---|---|---|---|---|
| OrthCaps-S | **105.5K** | 673.1 | **99.68** | **96.26** | **98.30** | **87.92** |
| Efficient-Caps | 162.4K | **631.1** | 99.58 | 93.12 | 97.46 | 81.51 |
| Capsnet | 8388 | 803.8K | 99.52 | 91.36 | 95.42 | 68.72 |
| Matrix-CapsNet with EM routing | 450K | 949.6 | 99.56 | 87.42 | 95.56 | 81.39 |
| AR CapsNet | 9.1M | 2562.7 | 99.46 | 85.98 | 96.47 | 85.39 |
| DA-CapsNet | 7M* | - | 99.53* | 94.82* | 98.26* | 85.47* |
| AA-Caps | 6.6M* | - | 99.34* | 91.23* | 89.72* | 79.41* |
| Baseline CNN | 4.6M | 1326.9 | 99.22 | 91.28 | 87.11 | 72.20 |

| Deep Networks | Param ↓ | FLOPS[M]↓ | CIFAR10 | CIFAR100 | MNIST | FashionMNIST |
|---|---|---|---|---|---|---|
| OrthCaps-D (simplified routing) | **164K** | 3156 | 89.09 | 67.43 | **99.72** | 93.19 |
| OrthCaps-D (complete routing) | 574K | 3345 | 90.56 | **70.56** | 99.59 | **94.60** |
| AR CapsNet(7 ensembled) | 6.3M | 16657.5 | 88.94 | 56.53 | 99.49 | 91.73 |
| Capsnet(7 ensembled) | 5.8M* | 5137.4* | 89.4* | - | - | - |
| Inverted Dot-Product | 1.4M | 5340.9 | 84.98 | 57.32 | 99.35 | 92.85 |
| RS-CapsNet | 5.0M* | - | 89.81* | 64.14* | - | 93.51* |
| DeepCaps | 13.5M | **2687** | **91.01** | 69.72 | 99.46 | 92.52 |
| ResNet-18[1] | 11.7M | 5578.8 | 95.10 | 77.60 | 99.29 | 93.32 |
| VGG-16[1] | 147.3M | 15143.1 | 93.57 | 73.10 | 99.21 | 92.21 |

As shown in Table (1), OrthCaps-S achieves superior efficiency with merely 105.5K parameters, outperforming CNN, CapsNet, and many variants. For instance, Efficient-Caps, a state-of-the-art model on efficiency, has about 50% more parameters. Despite its compact design, OrthCaps-S either outperforms or matches the performance of other capsule network designs across all four datasets. On the SVHN and CIFAR10, OrthCaps-S achieves accuracies of 96.26% and 87.92%, respectively, surpassing CapsNet which has 80 times more parameters. With a computational demand of 673.1M FLOPS, it's worth noting that the slight increase in FLOPS compared to Efficient-Caps is due to the additional computations from the pruned capsule layer and orthogonal transformations. Given the substantial decrease in parameter count and the enhanced accuracy of both networks, this FLOPS trade-off is warranted.

For OrthCaps-D, as illustrated in Table (1), it exhibits competitive performance with fewer parameters and less computational cost. On complex datasets, OrthCaps-D delivers compelling results using fewer parameters. Although convolution-based networks such as ResNet-18 and VGG-16 perform well on CIFAR10 and CIFAR100, OrthCaps-D offers competitive performance using just 1.41% and 0.11% of their parameters as well as 56% and 20.8% of their FLOPS, respectively. The efficiency of OrthCaps becomes evident when compared with DeepCaps. While DeepCaps achieves a 91.01% accuracy on CIFAR10, its significant parameter count of 13.42M highlights a compromise. It's noteworthy that both OrthCaps variants maintain high performance with fewer parameters.

## 4.3 ABLATION STUDY

**Orthogonal Self-Attention Routing**

Through a cross-comparison of accuracy (ACC) and frames-per-second (FPS), as shown in the Table (b) of Figure (5), we contrast attention routing with dynamic routing(Sabour et al., 2017) and compare sparse softmax with standard softmax. Attention routing consistently outperforms dynamic routing in both classification accuracy and processing speed, achieving a 25.8% speed enhancement on average. Even with a faster softmax, dynamic routing only reaches 1339 FPS, indicating its inherent computational inefficiencies. While the complexity of $\alpha$-entmax over softmax and the added computations from orthogonality lead to a slight decrease in speed compared to using the softmax, the trade-off brings a significant accuracy boost at a small reduction in FPS.

---

[1]https://github.com/kuangliu/pytorch-cifar

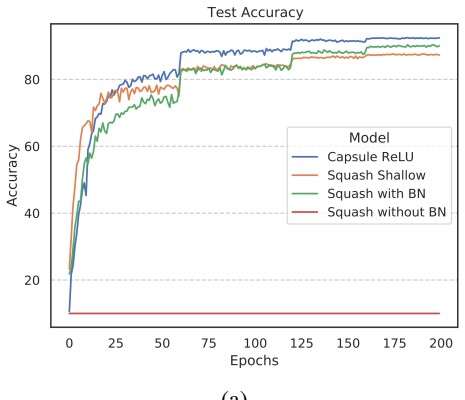

(a)

| Variants | FPS↑ | ACC↑ |
|---|---|---|
| Attention routing & $\alpha$-entmax & orthogonality | 1639 | 99.69 |
| Attention routing & softmax | 1785 | 99.62 |
| Dynamic routing & $\alpha$-entmax & orthogonality | 1232 | 99.51 |
| Dynamic routing & softmax | 1339 | 99.49 |

(b)

| Variant | Param[K]↓ | ACC↑ |
|---|---|---|
| OrthCaps-S with pruning | 105 | 99.69 |
| OrthCaps-S | 127 | 99.63 |
| Capsnet with pruning | 7492 | 99.45 |
| Capsnet | 8388 | 99.42 |

(c)

Figure 5: **Ablation study results. (a):** Test accuracy curve of different activation functions of OrthCaps-D and OrthCaps-S model on CIFAR-10. We train for 200 epochs with a learning rate of 0.001 and decayed the learning rate to $80\%$ of its original value at epochs 60, 120, and 160. **(b):** Comparison of Orthogonal sparse attention routing and dynamic routing algorithms on MNIST. We report the performance of OrthCaps-S trained 300 epochs. **(c):** CapsNets are compared with and without the pruning layer on the MNIST dataset, with the similarity threshold set to 0.7.

Overall, our attention routing combined with $\alpha$-entmax and orthogonality balances performance and computational efficiency.

**Pruned Capsule Layer**

Figure (1) illustrates that by integrating the pruned layer, the average capsule similarity decreases significantly due to redundant capsule elimination. Consequently, as the capsule count reduces, the dimensions of the associated prediction matrix diminish, thereby lowering the parameter count. This is proved in Table (c) of Figure(5), where the pruned version of OrthCaps-S has fewer parameters, reduced from 127K to 105K. Despite the reduction, performance is not compromised; in fact, the pruned model achieves an improved accuracy of 99.69%, compared to 99.63% for the unpruned version. When applying similar pruning to CapsNet, it yields an accuracy of 99.45% with 7492K parameters, which is fewer than its original 8388K parameters. Evidently, our pruning approach streamlines the model by eliminating redundant capsules and slightly enhances its performance.

Figure (6) illustrates the necessity of incorporating pruning with orthogonality. Capsule similarity is gauged with cosine similarity to measure the redundancy as mentioned above. As the network goes deeper, the dashed line (indicating pruning without orthogonality) shifts rightward, suggesting an increase in capsule similarity and redundancy. This shift proves that the alteration in capsule direction from the weight matrix reintroduces redundancy. In contrast, the solid line (indicating pruning with orthogonality) demonstrates consistently low capsule similarity. Even at 28 layers deep, the similarity remains below the 0.7 pruning threshold, affirming the efficacy of orthogonality in preserving capsule directions to maintain low inter-capsule correlations. The black dash-dot line denotes similarity without orthogonality and pruning, exhibiting the highest redundancy, further emphasizing the significance of our pruning approach.

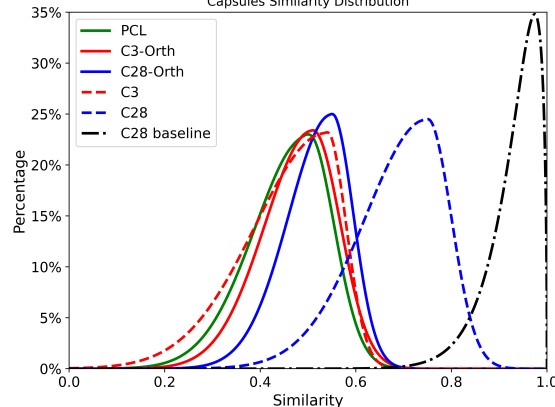

Figure 6: **Redundancy comparison between different pruning strategies. (The more to the left, the better.)** The x-axis shows capsule similarity; the y-axis indicates capsule count percentage. PCL, C3, and C28 mark the primary, third, and twenty-eighth capsule layers. Solid (C3-Orth, C28-Orth) and dashed lines contrast pruning with and without orthogonality; the dash-dot line shows no pruning or orthogonality.

**Capsule ReLU**

Figure (5)(a) presents the performance of various activation functions on CIFAR10. In shallow networks, squash converges faster, evident from its steep accuracy trajectory. However, in deeper

networks, squash without batch normalization struggles to learn useful information during training, leading to an accuracy near 10%. Introducing batch normalization to this deep squash model enhances its accuracy, underscoring the importance of batch normalization. In contrast, models employing Capsule ReLU as their activation function show quicker convergence during gradient optimization and achieve superior local minima.

## 4.4 ROBUSTNESS TO ADVERSARIAL ATTACKS

Capsule networks have demonstrated exceptional performance in terms of robustness (Hinton et al., 2018). Considering OrthCaps as it eliminates redundant capsules to suppress low $L_2$-norm capsules, which we consider as noise capsules (De Sousa Ribeiro et al., 2020). It can enhance better robustness against small perturbations. To evaluate this, we conduct a robustness comparison between OrthCaps, Capsule Networks, Orthogonal CNNs (OCNN) and 7-layer CNNs using the CIFAR10 dataset. We employ the Projected Gradient Descent (PGD) white-box attack method(Guo et al., 2019), setting the maximum iteration count at 40, step size at 0.01, and the maximum perturbation at 0.1. We assess the robustness using three metrics: attack time (AT), model query count (QC), and accuracy after attacks (Acc). As shown in Table (5), OrthCaps outperforms in all three metrics, confirming its superior handling of complex spatial structures. Specifically, OrthCaps requires 1.72 times more queries compared to CapsNet and exhibits a 9% higher accuracy after attacks, showing its robustness in image classification.

Table 4: Orthogonality of weight matrices in attention routing of SVHN dataset. $O$ decreases from 0.02 to 0.01 during training.

| EPOCH | ACC | $O \downarrow$ |
|---|---|---|
| 1 | 83.75 | 0.0236 |
| 10 | 98.58 | 0.0215 |
| 100 | 99.42 | 0.0153 |
| 300 | 99.56 | 0.0120 |

Table 5: Comparison of 7 ensembled OrthCaps, CapsNet, OCNN and baseline CNN under PGD attack. The CIFAR10 dataset is used without any data augmentation.

| Variants | AT(s) ↑ | QC[K] ↑ | Acc ↑ |
|---|---|---|---|
| OrthCaps | **345.92** | **69K** | **23.52%** |
| CapsNet | 198.93 | 48K | 14.62% |
| OCNN | 136.7 | 46K | - |
| baseline CNN | 16.65 | 10K | 0.35% |

## 4.5 ORTHOGONALITY

This experiment demonstrates the effectiveness of the HouseHolder orthogonalization method and its advantages over other orthogonalization methods. We define an orthogonality metric $O = \|K^T K - I\|$. In Table (4), $O$ decreases from 0.02 to 0.01 during training, substantiating the effectiveness of the orthogonalization method.

We further demonstrate Householder's role as a regularization technique for neural networks, detailed in Appendix A.3.2. In Figure (7), our method achieves better orthogonality and loss decay than OCNN (Wang et al., 2020). The baseline ResNet18, without any orthogonal regularization, is depicted by the blue line. while the green and red lines stand for OCNN and our method, respectively. Despite the decrease in orthogonality loss for OCNN throughout training, it remains almost 10 times higher compared to our Householder technique. The near-flat trajectory of the red line testifies to Householder's consistent orthogonality preservation across the training. Furthermore, our method registers a smaller loss than OCNN, due to its better training performance.

## 5 CONCLUSIONS AND FUTURE WORK

In this study, we have introduced a novel capsule network with orthogonal sparse attention routing. Specifically, Householder orthogonal decomposition is used to ensure strict matrix orthogonality in attention routing without additional penalty terms, and the capsule pruning layer introduces sparsity into routing, minimizing capsule redundancies. Our new activation function called Capsule ReLU mitigates the vanishing gradient problem. It has been shown in experiments that OrthCaps has lower parameters and reduces computational overhead, overcoming the challenges of computational expense and redundancy in dynamic routing. On image classification tasks, OrthCaps outperforms state-of-the-art methods, demonstrating improved robustness. This work paves the way for future research in capsule networks, and we look forward to further developments in this area.

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

# A APPENDIX

## A.1 SYMBOLS AND ABBREVIATION USED IN THIS PAPER

| Symbol | Description |
|---|---|
| OrthCaps | Orthogonal Capsule Network |
| OrthCaps-S | Shallow network variant of OrthCaps |
| OrthCaps-D | Deep network variant of OrthCaps |
| $x$ | Input image |
| $l$ | Layer index |
| $\Phi^l$ | Features from convolutional layers at level $l$ |
| $u_l$ | Capsules at layer $l$ |
| $v_{l+1}$ | Capsules at layer $l+1$ |
| $n$ | Capsule count in a given layer |
| $d$ | Capsule dimension |
| $W$ | Feature map width |
| $H$ | Feature map height |
| $B$ | Batch size |
| $u_{flat}$ | Flattened capsules |
| $u_{sorted}$ | Capsules sorted by their $L_2$-norm |
| $u_{pruned}$ | Pruned capsules |
| $M$ | Mask matrix for pruned capsule layer |
| $S$ | Cosine similarity matrix for pruned capsule layer |
| $\theta$ | Threshold for pruned capsule layer |
| $W_{ij}$ | Weight matrix for simplified attention routing |
| $Q, K, V$ | Attention routing components: Query, Key, Value |
| $W_Q, W_K, W_V$ | Weight matrices for Q, K, V |
| $C_{i,j}$ | Coefficient matrix for attention routing |
| $s_{i,j}$ | Votes for attention routing |
| $g$ | Activation function |
| $H$ | Householder matrix |
| $a_i$ | Unit vector in Householder matrix formulation |
| $b_i$ | Learnable vector in Householder matrix |

## A.2 HYPERPARAMETERS

| Hyperparameter | Value |
|---|---|
| Batchsize | 512 (4 paralleled) |
| Learning rate | 5e-3 |
| Weight decay | 5e-4 |
| Optimizer | AdamW |
| Scheduler | CosineAnnealingLR and 5-cycle linear warm-up |
| Epochs | 300 |
| Data augmentation | RandomHorizonFlip, RandonClip with padding of 4 |
| Dropout | 0.25 |
| $m^+$ | 0.9 |
| $m^-$ | 0.1 |
| $\lambda$ | 0.5 |
| $\theta$ | 0.7 |
| $d$ | 16 |

### A.3 HOUSEHOLDER ORTHOGONALIZATION

### A.3.1 PROOF OF LEMMA 2

**Proof:**

Let $W$ represent one of $W_Q, W_K, W_V$ as $W$ can be expressed as

$$W = H_0 H_1 \ldots H_{d-1} \tag{5}$$

where $H_i = I - 2a_i a_i^T$. We have

$$W^T W = H_{d-1}^T \ldots H_1^T H_0^T H_0 H_1 \ldots H_{d-1} \tag{6}$$

We demonstrate that $H_i$ is orthogonal, i.e. $H_i^T H_i = I$. This is obvious, as

$$\begin{aligned} H_i^T H_i &= (I - 2a_i a_i^T)^T (I - 2a_i a_i^T) \\ &= I - 4a_i a_i^T + 4a_i a_i^T = I \end{aligned} \tag{7}$$

Therefore, Equation (6) can be written as $W^T W = \underbrace{I \ldots I}_{d} = I$.

### A.3.2 HOUSEHOLDER AS A REGULARIZATION TECHNIQUE

We demonstrate Householder's role as a regularization technique for neural networks. For ResNet18, we flatten and concatenate convolutional kernels into a matrix $W$, and orthogonalize it to minimize off-diagonal elements, which reduces channel-wise filter similarity and redundancy. To quantify these properties, we used Guided Backpropagation to dynamically visualize the activations(Wang et al., 2020). Compared to directly computing the covariance matrix of convolutional kernels, The gradient-based covariance matrix offers a more comprehensive view of the dynamic behavior of filters. We define the gradients from Guided Backpropagation as $G$ and compute its gradient correlation matrix $corr(G)$ as:

$$\left(\mathrm{diag}(K_{GG})\right)^{-\frac{1}{2}} K_{GG} \left(\mathrm{diag}(K_{GG})\right)^{-\frac{1}{2}} \tag{8}$$

where $K_{GG} = \frac{1}{M}\left((G - \mathbb{E}[G])(G - \mathbb{E}[G])^T\right)$, $M$ is the number of channels. Figure (7) of the off-diagonal elements of $corr(G)$ shows a left-shifted distribution for the Householder-regularized model, confirming its effectiveness in enhancing filter diversity and reducing redundancy.

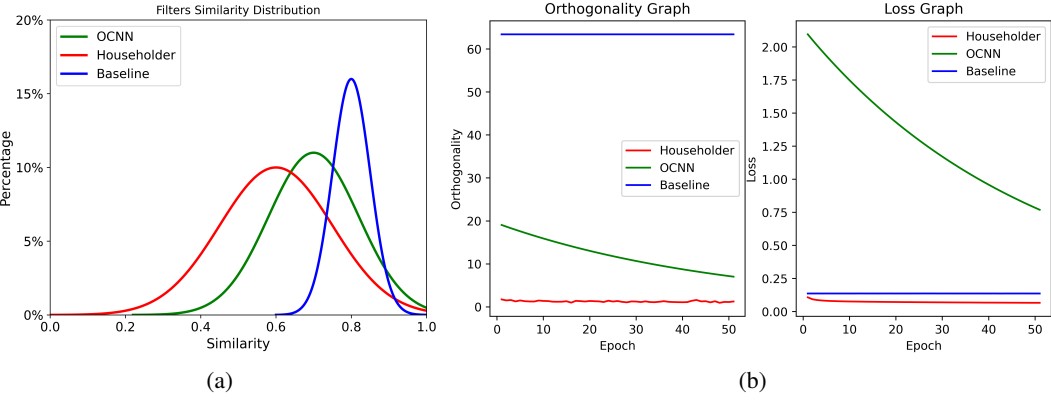

Figure 7: **(a):** The normalized histogram of pairwise filter similarities in standard ResNet34 with different regularizers. HouseHolder orthogonalization method shows the best performance of descending filter similarity. **(b):** Capsnet with different Orthogonal regularization on MNIST dataset. Our HouseHolder orthogonalization method reaches better orthogonality and loss decay.

### A.4 CAPSULE NETWORK

#### A.4.1 DYNAMIC ROUTING

Algorithm 2 describes the dynamic routing algorithm. This algorithm allows lower-level capsule output vectors to be allocated to higher-level capsules based on their similarity, thereby achieving an adaptive feature combination. However, as evident from $\sum_i c_{ij} \hat{u}_{j|i}$, each higher-level capsule is a weighted sum of lower-level capsules. The higher-level capsules are fully connected with the lower level. Furthermore, the routing algorithm fundamentally serves as an unsupervised clustering process for capsules, requiring $r$ iterations to converge the coupling coefficients $c$. It's crucial to strike a balance in choosing $r$: an inadequate number of iterations may hinder convergence of $c$, impairing routing efficacy, while an excessive count increases computational demands. In

---

**Algorithm 2** Dynamic Routing

---

1: **procedure** ROUTING($\hat{u}_{j|i}, r, l$)
2:     **for** all capsule $i$ in layer $l$ and capsule $j$ in layer $(l+1)$ **do** $b_{ij} \leftarrow 0$
3:     **for** $T$ iterations **do**
4:         **for** all capsule $i$ in layer $l$ **do** $c_i \leftarrow \text{softmax}(b_i)$
5:         **for** all capsule $j$ in layer $(l+1)$ **do** $s_j \leftarrow \sum_i c_{ij} \hat{u}_{j|i}$
6:         **for** all capsule $j$ in layer $(l+1)$ **do** $v_j \leftarrow \text{squash}(s_j)$
7:         **for** all capsule $i$ in layer $l$ and capsule $j$ in layer $(l+1)$ **do** $b_{ij} \leftarrow b_{ij} + \hat{u}_{j|i} \cdot v_j$
8:     **return** $v_j$

---

Conclusion, it is crucial to introduce a straightforward, iterative-free routing algorithm.

#### A.4.2 SQUASH ACTIVATION FUNCTION

Figure(8) displays the functions and their derivatives for both sigmoid and squash. The x-axis represents the $L_2$-norm of the vote from routing, serving as the function input, while the y-axis denotes the function values and their respective derivatives.

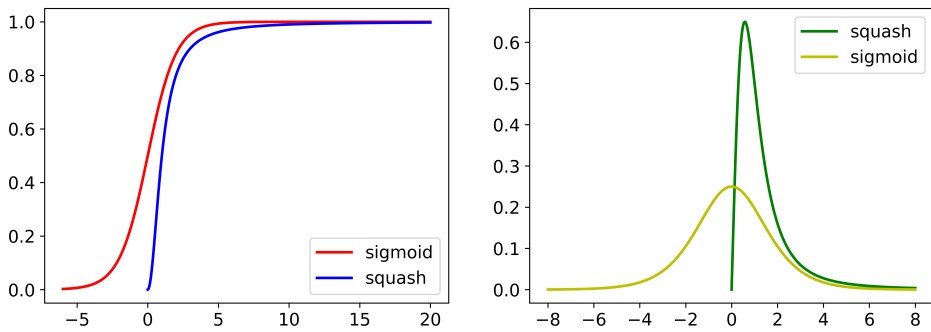

Figure 8: Comparison of function and derivative figure of sigmoid and squash. **Left:** Function figure of sigmoid and squash. **Right:** Derivative figure of sigmoid and squash.

The Squash function is defined as:

$$v_j = \frac{||s_j||^2}{1 + ||s_j||^2} \frac{s_j}{||s_j||}$$

where $v_j$ is the output vector, $s_j$ is the input vector, and $||s_j||$ represents the L2 norm of the input vector.

When the norm $||s_j||$ of the input vector $s_j$ approaches zero or infinity, the output $v_j$ of the Squash function tends to be zero, due to the dominance of the denominator $1 + ||s_j||$ in the former and

the normalization by the vector magnitude $||s_j||^2$ in the latter. For intermediate magnitudes, the function undergoes a rapid transition. When $||s_j||$ is near zero, the output $v_j$ of the Squash function changes relatively rapidly, which is not conducive to gradient optimization and leads to unstable training. The derivative approaches zero when the norm $||s_j||$ of the input vector $s_j$ approaches zero or infinity, leading to gradient vanishing issues.

Thus, it is necessary to design a new activation function to replace the Squash function for the deep Capsule Network.

