# OpenReview forum: "OrthCaps: An Orthogonal CapsNet with Sparse Attention Routing and Pruning"
_ICLR.cc/2024/Conference — ICLR 2024 Conference Withdrawn Submission_

### Official Review · Reviewer_1dRT · 2023-10-24

**Soundness:** 2 fair
**Presentation:** 2 fair
**Contribution:** 3 good
**Rating:** 5
**Confidence:** 4

**Summary:**

This paper presents two capsule network architectures, denoted OrthCaps-D and OrthCaps-S. Householder orthogonal decomposition is used to reduce the redundancy. Moreover, the Capsule ReLU activation function is introduced. The experiments show comparable or better results than related works.

**Strengths:**

1. The work is original and relevant to the community.

2. The results show that the proposed architectures are comparable or better than the related works.

**Weaknesses:**

Some parts are not described in sufficient detail. Please refer to the questions below.

**Questions:**

1. The GitHub link provided in the paper (https://github.com/ornamentt/Orthogonal-Capsnet) points to an empty repository. Is it possible to make the code available for reviewers’ inspection?

2. In Section 1, please provide all the detailed setup necessary for generating the results in Figure 1.

3. In Section 3, the architecture of the Capsule Block is not clear. Please describe in detail how the ConvCaps are connected, how many Capsule Blocks are there, etc.

4. OrthCaps-D and OrthCaps-S are two architectures with specific structures and (hyper-)parameters. Please describe more in detail the design decisions made to develop these architectures.

5. In Section 4.3, some ablation studies (results shown in Tables b and c of Figure 5) are conducted only on the MNIST dataset. It is recommended to conduct these experiments on more complex benchmarks (e.g., CIFAR10, CIFAR100) to investigate whether these findings are consistent with other benchmarks and to better appreciate the accuracy variation.

---

### Official Review · Reviewer_EBRe · 2023-10-28

**Soundness:** 3 good
**Presentation:** 3 good
**Contribution:** 3 good
**Rating:** 6
**Confidence:** 5

**Summary:**

The paper proposed a new process of routing in capsule neural networks (henceforth CapsNets) that incorporates orthogonalisation to prevent feature overlap, introduces a capsule relu with batchnorm, and pruning. The gist is to reduce the number of parameters and develop variations that are shallower and deeper. Obviously shallow and deeper in CapsNets shall not be conflated with Transformers (or even CNNs) as the number of parameters is few. Since Dynamic routing (about 8M parameters) most ConvNets flavours have been considerably smaller, while performing well across SmallNorb, SVHN, FashionMnist, Cifar 10, Affnist etc. \
The authors have provided some appropriate references but have also missed some papers that are still SOTA, e.g. Introducing Routing Uncertainty in Capsule Networks (98.6% accuracy with 140K parameters and 97.8% with 60K). Also the authors mention Affnist but they never provided any results (as far as I can tell - appendix does not have them either). Overall I think this is a good paper and am not invested in SOTA performance as I think CapsNets have merits beyond what is attributed to them (largely because of how transformers scale compared to CapsNets).

**Strengths:**

The paper offers a nice formulation of a non-iterative orthogonalised pruned variation of capsules that works well with a Relu+batchnorm activation function. None of the three components is novel in that regard, but one would argue what is novel in the 1000s of papers that come out based on transformers (off topic). The performance presented (although not SOTA on SmallNorb (see page 6 in that NeurIPS paper: https://proceedings.neurips.cc/paper/2020/file/47fd3c87f42f55d4b233417d49c34783-Paper.pdf), is reminiscent of an overall positive trajectory that is theoretically sound. For instance, in the pruning context, the holy grail is achieving sparsity (especially in a structured setting) without compromising performance.  \
To be personally convinced I would like to see some performance in Affnist settings, although Smallnorb is a good proxy overall and plays on CapsNets strengths.

**Weaknesses:**

Firstly, please revisit some of the SOTA claims - in table 1 the authors have various models of various sizes so not including the above paper I mentioned (uncertainty routing as well as VB routing  by Ribeiro et al. that the authors already cited) seems a bit odd. Correct me if I am wrong but as far as I am aware these two papers are still the SOTA on SmallNorb and FashionMnist (VB routing achieved 94.8% on Fashionmnist with 172K parameters compared with 94.6% and 574K parameters in this paper), and perhaps affnist and multimnist? Nevertheless, I think it mostly matters for the results presented in table 1, as well as affnist that is mentioned but never provided.

**Questions:**

I think my questions are sort of evident from what I have written above.

a) Can you please realign your results concerning other SOTA papers? \
b) Wasn't Relu+Batchnorm for capsules first mentioned in the VB routing section 4 page 5? \(https://ojs.aaai.org/index.php/AAAI/article/view/5785) \
c) can you please expand upon your conclusion that the results in figure 5 point to the vanishing gradient issue? the squash+batchnorm vs relu+batchnorm shows some improvement but is that really due to vanishing gradients? Unless that is used in the wider context of capsules dying? check this paper:
Mitterreiter, Matthias, Marcel Koch, Joachim Giesen, and Sören Laue. "Why Capsule Neural Networks Do Not Scale: Challenging the Dynamic Parse-Tree Assumption." arXiv preprint arXiv:2301.01583 (2023). \

---

### Official Review · Reviewer_g9dd · 2023-10-30

**Soundness:** 2 fair
**Presentation:** 3 good
**Contribution:** 2 fair
**Rating:** 6
**Confidence:** 5

**Summary:**

The paper proposes a new capsule-based framework called OrthCaps to reduce the redundancy of capsule networks. The contributions includes: (1) the pruned layer which deactivates the capsules with high cosine similarity; (2) the orthogonal sparse attention routing to replace the dynamic routing for efficiency; (3) the Capsule ReLU to address vanishing gradient. Experiments on MNIST, SVHN, smallNORB and CIFAR10 validate the effectiveness of the proposed method.

**Strengths:**

1.	The paper focuses on the redundancy issues of capsule networks, which is a highly valuable problem. The proposed sparse regularizations also demonstrate feasibility.
2.	With fewer parameters, the proposed method achieves impressive performance on CIFAR10 and CIFAR100 datasets.
3.	The paper is well-written and the contributions are illustrated clearly.

**Weaknesses:**

1.	The superiority of alpha-entmax and orthogonality is not validated. As shown in Table(b) of Figure (5), the performance with alpha-entmax and orthogonality is improved very slightly: 0.07 and 0.02, while with >100 FPS slower. To highlight the effectiveness of sparse, the authors are suggested to show the robustness metrics (as in Table 5) at the same time.
2.	With more params, OrthCaps-D performs worse than OrthCaps-S on MNIST. Is that means that the performance on MNIST is saturated? Besides, OrthCaps-D (simplified) shows comparable performance with OrthCaps-S and with similar params, what is the motivation to design OrthCaps-S?
3.	As illustrated in the paper, reducing the inefficiencies and redundancy will help the network stacks deeper, the authors are suggested to show the results of a more deeper architectures (for e.g.  reaching the Param of resnet 18) with more competitive performance.

**Questions:**

1.	With threshold $\theta$ for selecting, what is the ‘sorted’ for? (Line 3 in Algorithm 1)
2.	How is the threshold $\theta$ settled? Does the choose of $\theta$ matters?
3.	Without the orthogonalization (HouseHolder orthogonalization method), is the orthogonal sparse attention degrades to the attention in the common transformers?

If the weakness and questions are well addressed, I will improve my rating to accept level.

---

### Official Review · Reviewer_TYvA · 2023-10-30

**Soundness:** 2 fair
**Presentation:** 1 poor
**Contribution:** 2 fair
**Rating:** 1
**Confidence:** 2

**Summary:**

This paper proposes the Orthogonal Capsule Network (OrthCaps) to reduce redundancy in capsule networks, improving routing performance and decreasing parameter count. 1) A pruned capsule layer is used to remove redundant capsules. 2) Householder parameterization is used to make the Q/K/V matrices orthogonal. 3) A capsule ReLU is proposed. Reasonable results are reported.

**Strengths:**

1. The idea of utilizing orthogonality is interesting.
2. Usage of Householder parameterization is natural and easy to understand.

**Weaknesses:**

1. This paper is very hard to follow. I got completely lost even reading the Introduction. I would suggest the authors completely rewrite the paper to make it readable.
1.1 It is mentioned that attention routing is used. But how does it eliminate the need for iteration?
1.2 What is the fully-connected problem and how is that solved by sparsemax-based self-attention?
1.3 Why is a simplified attention-routing mechanism used with OrthCaps-S and what is it?
1.4 What are capsule vector direction shifts and how are they solved with orthogonality? Why does preserving the direction of capsule vectors mitigate feature interference?
1.5 What is the motivation of  Capsule ReLU? Why is that used with OrthCaps-D but not OrthCaps-S?

2. Why is capsule ReLU designed so? What is the motivation for using BN on a scalar (||s_j||^2)? It is very unnatural, so I would expect a clear motivation.

I rate a strong reject mostly because the writing quality of this paper makes it difficult for me to accurately understand the essence of the proposed method, thus preventing me from evaluating its technical correctness and significance.

Of course, as I am not an expert in capsule networks, it may be due to my lack of relevant knowledge that I cannot understand this paper. If the writing of this paper is not the issue, that is, if AC and other reviewers can understand this paper without difficulty, please let me know, and I am willing to modify my rating.

**Questions:**

See the weakness part.

---

### Official Review · Reviewer_ETfq · 2023-11-01

**Soundness:** 2 fair
**Presentation:** 2 fair
**Contribution:** 2 fair
**Rating:** 3
**Confidence:** 5

**Summary:**

The authors propose a novel orthogonal sparse attention routing mechanism to address the problem of high computational cost and parameter counting caused by redundancy. In addition, a pruning capsule layer is placed to alleviate capsule redundancy, and a new activation function capsule ReLU is proposed for deep capsule network to solve the problem of gradient disappearance.

**Strengths:**

1.It is the first time that orthogonality is introduced into capsule networks.
2.The proposed approach addresses the redundancy problem from both the primary capsule and the routing mechanism.

**Weaknesses:**

1. The introduction to orthogonality in Part 2 could be more detailed.
2. No details on how the capsule blocks are connected to each other.
3. The fourth line of Algorithm 1 does not state why the flatten operation is performed.
4.The presentation of the α-enmax function is not clear.
5. Eq. (4) does not specify why BatchNorm is used for scalars (L2-norm of sj).
6. The proposed method was tested on relatively small datasets, so that the effectiveness of the method was not well evaluated.

**Questions:**

1. The introduction to orthogonality in Part 2 could be more detailed.
2. No details on how the capsule blocks are connected to each other.
3. The fourth line of Algorithm 1 does not state why the flatten operation is performed.
4.The presentation of the α-enmax function is not clear.
5. Eq. (4) does not specify why BatchNorm is used for scalars (L2-norm of sj).
6. The proposed method was tested on relatively small datasets, so that the effectiveness of the method was not well evaluated.